# Knowledge, Attitude, and Practice Regarding Antibiotic Use and Resistance for Upper Respiratory Tract Infections among the Population Attending a Mass Gathering in Central India: A Cross-Sectional Study

**DOI:** 10.3390/antibiotics11111473

**Published:** 2022-10-25

**Authors:** Ngoc V. Nguyen, Yogyata Marothi, Megha Sharma

**Affiliations:** 1Department of Global Public Health—Health Systems and Policy: Medicines, Karolinska Institutet, 17177 Stockholm, Sweden; 2Clinical Epidemiology Division, Department of Medicine Solna, Karolinska Institutet, 17177 Stockholm, Sweden; 3Department of Microbiology, R. D. Gardi Medical College, Ujjain 456006, India; 4Department of Pharmacology, R. D. Gardi Medical College, Ujjain 456006, India

**Keywords:** antibiotic use, antibiotics, antibiotic resistance, community, knowledge, attitude, practice, URTI, mass gathering

## Abstract

Background: Good knowledge and appropriate attitude and practice about antibiotic use and resistance among the general population, are significant contributors to minimize the development of antibiotic resistance. We aimed to study the knowledge, attitude, and practice (KAP) regarding antibiotic use, resistance to upper respiratory tract infections (URTI), and associated factors with KAP, among the population attending a mass gathering in India. Methods: A cross-sectional study was conducted in 2016 during a mass gathering held in Ujjain city of Central India. A self-administered, pre-tested questionnaire consisting of 28 questions was used to collect data on demographic characteristics, KAP related to antibiotic use, resistance, and URTI. Descriptive analyses were used to describe participants’ responses. Participants were divided into poor or good knowledge and appropriate or inappropriate groups of attitude and practice. Multivariable logistic regression models were used to examine the associations between demographic characteristics, URTI knowledge, and each domain of KAP. All statistical analyses were performed using Stata 16.0. The significance level was set at 5%. Results: A total of 1915 participants consented to participate (response rate 92.7%) with a mean age of 39.3 (±14.7). Complete data on socio-demographics were available for 1619 participants. Of these, 59% were male, and 61% had an education level below high school. Eighty-nine percent of participants had poor knowledge about URTI. A majority of the respondents defined antibiotics incorrectly (93%) and were classified as having poor knowledge (97%). Most of the participants (63%) could not mention any contributors to the irrational use of antibiotics. Appropriate attitudes were observed in 40% of participants, 87% denied to comply with the prescribed course of antibiotics and 88.5% had inappropriate responses for practice. Age of the respondent, sex, education, occupation, and knowledge about URTI, were the main factors associated with the KAP. Conclusions: KAP about antibiotic use and resistance among the general population in India was poor. Knowledge about URTI is strongly associated with KAP. Community interventions, i.e., educational campaigns, should be designed and implemented promptly considering the differences in demographics of the target audience.

## 1. Introduction

Antibiotic resistance (ABR) represents one of the biggest public health threats of the 21st century [1]. The problem respects no age group and no geographical boundaries and is a global health crisis. ABR is triggered by the inappropriate and irrational use of antibiotics [1]. ABR can cause treatment failure, prolonged hospital stay, increased healthcare costs, and more severe infectious conditions, that can lead to mortality [1].

The burden of infectious disease is high in low- and middle-income countries (LMIC) and is highest in India [2,3,4]. In 2010, India had one of the most challenging situations of ABR and was the largest consumer of antibiotics worldwide (12.9 × 10^9^ units, 10.7 units/person) [2,4]. The irrational use of antibiotics in LMIC may arise from a complex interaction between prescriber and patient, giving rise to prescriber-related factors such as the lack of policies to regulate antibiotic prescribing and dispensing, poor prescribers’ knowledge, and the easy procurement of antibiotics over the counter (OTC), without prescription [5,6]. Importantly, patient-related factors such as self-medication, inappropriate knowledge, attitude, and practice (KAP) regarding antibiotics, and inappropriate expectations from antibiotics, are also significant contributors to the development and spread of ABR [5,6]. The population’s knowledge, perception, and attitudes are crucial in several public health issues. The progression of COVID-19, a respiratory tract infection, as a pandemic is the most relevant example where the lack of knowledge about how to prevent the spread of infection was one of the major factors that claimed several lives [7]. Moreover, the improved knowledge played a crucial role in minimizing the spread as well. It is noteworthy that the consequences of overuse of antibiotics during the COVID-19 pandemic period are yet to be studied. Moreover, a high prevalence of broad-spectrum antibiotic-resistant pathogens has been observed in many regions in India [2,8,9].

The population’s knowledge and attitude about antibiotics play an important role in their use since antibiotics can also be demanded or purchased OTC at pharmacies in LMIC. This eventually relates to the development of bacterial resistance and the spread of infections [10,11,12,13,14,15]. Therefore, public involvement is crucial for the containment of ABR. The World Health Organization (WHO) recommends public involvement via three key issues: improving access to medical facilities; decreasing the unnecessary use of antibiotics; and complying with treatment and not sharing antibiotics, or saving leftover antibiotics for future use [16].

Upper respiratory tract infections (URTI) are among the most common infectious diseases and one of the top three diagnoses in outpatient settings across the country [17]. Acute URTI remains the clinical condition for which antibiotics are most commonly prescribed [18,19]. However, most of these prescriptions are classic examples of the excessive use of antibiotics since the leading cause of URTI is viruses, and less than 10% of URTI cases are caused by bacteria, which require antibiotic treatment [17]. Additionally, according to a systematic review, URTI was the most common reason for self-medication with antibiotics in the southeast Asian region, including India [20]. Therefore, the inappropriate use of antibiotics for URTI could substantially contribute to the increasing spread of ABR. Furthermore, the public’s knowledge about URTI can influence a major part of the total use of antibiotics.

Educational interventions for the general population regarding antibiotic use and resistance have also been highly recommended by the WHO, and also included in the National Action Plan on Antimicrobial Resistance (2017) by the Government of India [21,22]. Information on the KAP regarding antibiotic use and ABR among the general population will facilitate to develop tailored and effective educational campaigns. Cross-sectional studies have assessed the KAP of antibiotic use among the general population in high-income countries [23,24,25], but very few such studies have been conducted in India [26]. Considering the large population and high burden of infectious diseases and ABR in India, there is an urgent need to understand the KAP related to antibiotic use and resistance among the general population, especially among the persons attending large mass gatherings such as (*Hajj* and *Kumbh Mela*) since these events could be a potential risk for the transmission of communicable diseases [27,28,29,30]. The present study primarily aims to assess the KAP about antibiotic use and ABR among the general population attending a mass gathering in India. The secondary aim was to explore the associations of KAP scores with the socio-demographic characteristics, the extent of knowledge about URTI, antibiotic use and ABR.

## 2. Methods

### 2.1. Study Design and Population

A descriptive cross-sectional study was conducted during a mass gathering (22 April 2016 and 21 May 2016) held in the Ujjain city of Madhya Pradesh, India. Adult participants (aged ≥ 18) who consented to participate in the study were included in the study. The gathering is known as the *Kumbh Mela* (*Kumbh* = Pot, *Mela* = fair), or *Simhastha Mela*. The *Kumbh Mela* is held for a month every 12th year in each of the four Indian cities, namely Haridwar, Ujjain, Nashik, and Prayagraj. This event is recognized as the world’s biggest mass gathering and usually attracts millions of pilgrims coming from different geographical regions in India and abroad [29].

### 2.2. Data Collection Tools and Methods

Data collection tool: a data collection questionnaire was developed by an expert panel comprised of a clinician, a microbiologist, and a pharmacologist. The questionnaire was specifically focused on the study aims and tailored contextually to fit local situations. The questions included were based on previous studies conducted on a similar topic in India, Kuwait, Malaysia, Ghana, and Sweden [25,26,31,32,33], and specific situations in India (for example the question on the red line in medication strips). A majority of questions were derived from universally accepted definitions to assess the KAP regarding antibiotic use and resistance (e.g., the definition of antibiotics, the consequences of irrational use of antibiotics, antibiotic use without prescription). The questionnaire was formulated in the Hindi language. Two proficient English-Hindi translators other than members of the expert panel independently translated the questionnaire into English and back to Hindi. The final Hindi version was compared with the original version of the questionnaire in the presence of the expert panel to reach a consensus on the reliability of the translation.

The questionnaire was then pilot-tested twice among the local community in a total of 22 participants (10 in the first round and 12 in the second round, data not included in the present communication) for its clarity and understanding. A few modifications were made after the pilot test by the expert panel.

The final questionnaire comprised of five sections with a total of 30 questions (Appendix A). Section 1 was information on socio-demographic characteristics (seven questions). Section 2 consisted of nine questions on URTIs, including one sub-question. Section 3, Section 4 and Section 5 included 14 questions (one sub-question) to assess participants’ KAP (four for knowledge, four for attitudes, and five for practice). A majority of questions (28/30 questions) were in multiple-choice format to overcome time constraints during the mass gathering. Two open-ended questions were: year of birth, and the definition of antibiotics.

Sampling: surveyors were trained for data collection and were allocated to different locations in rotation. It was a self-administrated questionnaire, but the surveyors were trained to fill the questionnaires for those participants who had difficulties reading and writing. The convenient sampling method was applied for data collection. This was deemed as the only possible sampling technique due to the nature of the mass gathering that included numerous flows of the attendees. The background and aim of the study were explained to the pilgrims using an information sheet, and the participants were asked for written consent before participation. For the participants who were unable to read and/or write and sign, oral informed consent was obtained, and a trained surveyor filled out the questionnaire according to the verbal responses received from the participants.

### 2.3. Outcome Assessment

Briefly, participants’ knowledge about antibiotics was assessed using questions on the definition, side effects, the efficacy of antibiotics, and the consequences of the irrational use of antibiotics. Participants’ attitudes were assessed based on their views on the future use of antibiotics, the causes of irrational use, and their expected behaviors if not prescribed antibiotics. We evaluated participants’ practice using questions on their awareness of the red line, antibiotic use without prescription, the use of leftover antibiotics, and antibiotic purchase behaviors. Further details can be found in Appendix A. The term “informal doctors” used in the present study refers to unlicensed, unregistered, and unqualified medical practitioners in India [34]. There is a ‘Red Line Campaign’ where the red line on the medicine strip indicates that those medicines must be used only when prescribed.

Scoring for outcome variables: for each question assessing the KAP, the appropriate or correct responses were given a score of one point, and incorrect or inappropriate responses were scored zero. The total score range of the knowledge, attitude, and practice parts was 0–4, 0–4, and 0–5, respectively. The cutoff of 2 points (>2 and ≤2) was selected to classify the participants into good (>2) or poor (≤2) knowledge, appropriate (>2) or inappropriate (≤2) attitude, and appropriate (>2) or inappropriate (≤2) practices, respectively [25,35].

### 2.4. Statistical Analysis

As per our aim, to separately estimate the proportions of individuals with good knowledge, appropriate attitude, and appropriate practice, the following formula was used for sample size calculation: $n\geq \frac{z^{2}P\left(1-P\right)}{\;\varepsilon^{2}}$ where *n* is the minimum needed sample size; *z* is 1.96 for a 95% confidence interval; *ε* is 5%, which is the margin of error; and *P* is the estimated proportion of good knowledge, appropriate attitude, and appropriate practice in previous literature. As previous studies have used different instruments to assess the KAP, and we had three different proportions of interest, we decided to use 0.5 as the p to maximize our sample size. Assuming a response rate of 50% due to time constraints during the mass gathering, a larger sample size of 768 individuals was considered.

A descriptive analysis was performed where the frequencies and percentages were reported for categorical variables. Means and standard deviations (SD) or median and interquartile ranges were reported for continuous variables. Multivariable logistic regression analyses were used to investigate the associations between socio-demographic characteristics, knowledge about URTI, and the KAP regarding antibiotic use and resistance. We analyzed knowledge, attitudes, and practice separately as dependent variables, which were coded as follows: good knowledge (1 = yes, 0 = no), appropriate attitudes (1 = yes, 0 = no), and appropriate practice (1 = yes, 0 = no). The independent variables were similar in each of the three models, including sex, age group, education, occupation, family type, place of residence, and knowledge about URTI.

Data entry was done using the EPI Info (version 3.1) and all statistical analyses were performed using the STATA (Version 16.1) with a significance level of 5%. Although the participants were encouraged to respond to all questions, some missing values were observed while analyzing data. A majority of missing values were observed in the socio-demographic section (15%) and the minimum in the practice part (0.2%). When analyzing the responses, participants with missing values on any variables of a specific section were excluded from the analysis.

## 3. Results

### 3.1. Socio-Demographic Characteristics

Overall, 2065 pilgrims were approached and invited to participate in the study. Of those, 1915 pilgrims consented to participate and fill out the questionnaire, giving a response rate of 92.7%. Complete data on socio-demographics were available for 1619 participants. The mean age was 39.0 (± 15.3), and the majority were 18 to 40 years old (65%). Of the 1619 respondents, 59% were male, and 52% had an education less than high school. Table 1 summarizes the socio-demographic characteristics of study participants.

### 3.2. Participants’ History and Knowledge of URTI

Of the 1915 respondents, 902 (47.1%) reported having a history of URTI. Among these participants, 96% did not know whether they had been prescribed an antibiotic or not for the last URTI episodes, and 75% believed they got the disease from the environment. Data on knowledge about URTI were available for 1868 participants. The median URTI knowledge score was 1 (range 0–4, IQR 0–2). All knowledge questions had a proportion of more than half incorrect responses. The question on URTI symptoms had the highest percentage of incorrect answers (94%), while only 6% of the participants could identify the typical symptoms of URTI. Less than 1% knew that URTI is caused by an organism or a germ, and more than 90% thought that weather change is a cause of URTI. Participants performed best in the question: “Is URTI communicable?” where nearly half gave a correct response. Consequently, poor knowledge (total score of 0–2) was observed in 89% of the participants (Table 2).

Data on antibiotic use for URTI were available for 1763 participants. Among them, 23% responded that antibiotics are not necessary to treat URTI. Prescription compliance was reported as the most common reason for use of antibiotics during URTI (49%). Among those who thought antibiotics are not necessary, the most commonly reported reason was the self-limiting nature of URTI (49%).

### 3.3. Knowledge about Antibiotic Use and ABR

Table 3 presents the findings from the assessment of participants’ knowledge about antibiotic use and resistance. A set of four questions was used with a total score range of 0–4. Overall, the level of knowledge was relatively poor with a median score of 0 (0–1), resulting in only 3% of participants being classified as having good knowledge (IQR 3–4). A majority of the participants responded incorrectly to all four questions (≥78%). Seven percent of the participants could define antibiotics correctly, and 11% could recognize a few potential consequences of irrational use of antibiotics, including ABR development, more severe disease, and treatment problems in the future. Twenty-two percent of the participants stated that antibiotics are losing their efficacy towards bacteria.

### 3.4. Attitude towards Antibiotic Use

Table 4 presents the responses of 1662 participants to the questions related to attitude. The median score for the attitude part was 2 (IQR 2–3). The majority did not know that antibiotics should be preserved for future use. More than 80% of participants said that they do not pressurize healthcare workers to prescribe antibiotics or visit another healthcare worker if antibiotics are not prescribed by the first healthcare worker (appropriate attitude). Overall, an appropriate attitude towards antibiotics was documented in 40% of participants. When asked: “Who do you think is responsible for the irrational use of antibiotics?” 36% gave responses that included allopathic doctors, local non-allopathic practitioners, patients, medical shops/pharmacies, pharmaceutical companies, and informal doctors (unlicensed doctors). According to participants’ views, registered doctors (30%) and informal doctors (15%) were the most common groups of people held responsible for ABR.

### 3.5. Practice Regarding Antibiotic Use and ABR

Participants’ practice regarding antibiotic use and ABR was evaluated using five questions and the results are illustrated in Table 5, with available data in 1911 participants. In three out of five questions, more than 80% of participants reflected inappropriate practices. Most participants (81%) did not know whether they had used antibiotics or not. Almost all participants (98%) did not pay attention to the red line on the medication strip/pack, which indicates prescription medication. Eighty-seven percent stated that they would not take the entire course but rather stop taking medicines when some symptomatic relief is achieved. Nearly half of the participants (48%) stated that they would buy the complete course of prescribed antibiotics. In this part, the question on leftover antibiotics had the highest proportion of appropriate responses with almost all participants saying that they would not use leftover antibiotics or would not fill the same prescription even if similar symptoms appear. In summary, the median score of the practice part was 2 (IQR: 1–2), and 89% of participants had inappropriate practice regarding antibiotic use.

Forty-seven participants provided information about the reason for buying antibiotics OTC, and the most common reason was the lack of money to pay a visit to healthcare (47%). Perceived symptomatic relief was the most common reason behind non-adherence to the instruction for the administration of the entire course of antibiotics (43%).

### 3.6. Factors Associated with KAP about Antibiotic Use and ABR

We performed multivariable logistic regression analyses to determine factors associated with participants’ knowledge, attitude, and practice about antibiotic use and ABR. Regarding knowledge, females (adjusted OR—aOR 2.26, 95%CI 1.05–4.85), above high school education level (aOR 24.16, 95%CI 9.77–59.73), and better URTI knowledge (aOR 2.90, 95%CI 1.35–6.22) were significantly associated with better knowledge about antibiotics, whereas the extended family group (aOR 0.32, 95%CI 0.14–0.72) was associated with poor knowledge. No association was found for age, occupation, and place of residence. For attitudes, age group 41–64 (aOR 0.72, 95%CI 0.55–0.95), age group ≥65 (aOR 0.54, 95%CI 0.35–0.86), female (aOR 0.65, 95%CI 0.49–0.86), and participants of extended family (aOR 0.51, 95%CI 0.32–0.83) were associated with lower odds of appropriate attitudes. However, living in the city (aOR 1.28, 95%CI 1.01–1.63) and better knowledge about URTI (aOR 5.85, 95%CI 3.94–8.68) were linked to appropriate attitudes. Factors associated with appropriate practice regarding antibiotic use and resistance included: having a job (aOR 1.80, 95%CI 1.17–2.78), above high school education (aOR 1.92, 95%CI 1.22–3.02), living in the city (aOR 1.56, 95%CI 1.11–2.20), and better knowledge about URTI (aOR 3.56, 95%CI 2.41–5.28). Detailed findings from multivariable analyses are presented in Table 6.

## 4. Discussion

The present study highlights that the overall KAP regarding antibiotic use and resistance among the participants attending the mass gathering, was poor. Positive associations were found between socio-demographic characteristics: female sex, higher education level, younger age, having a job; and the KAP about antibiotic use and resistance. We expect that our results would serve as evidence to develop and organize educational campaigns targeting the general population to minimize the OTC purchase of antibiotics and thus slow down and combat ABR. Since there is an absolute lack of similar studies, we have compared our findings with the available, most comparable global studies.

Poor knowledge about antibiotics was significantly prevalent in our study participants (97%). This figure is much higher than the ones reported in comparable studies conducted in Kuwait and Ghana [25,32]. The Kuwaiti study was a cross-sectional survey conducted among 770 individuals and found that 47% of participants had a low level of knowledge [25]. A population-based survey in Ghana on 632 respondents reported that 25% of respondents had poor knowledge about antibiotic use and resistance [32]. It is worth mentioning that the number of questions in the knowledge part and the scoring method in each study were different. However, the questions in all studies were focused on the action and safety of the use of antibiotics and the main cause of the development of ABR. These two studies grouped participants’ knowledge into three levels (low, moderate, and high) instead of two in our study (poor and good) [25,32]. In the present study, 93% of the participants could not define antibiotics and did not know that these are used to treat bacterial infections only. This is higher than the result of a Malaysian study where 83% of participants could not identify that antibiotics cannot treat viral infections [31].

Poor knowledge is one of the potential contributors to the development of antibiotic misuse and ABR. Our participants’ knowledge about ABR was also poor with the majority not knowing any of the consequences of the irrational use of antibiotics and that antibiotic efficacy is threatened due to ABR. One possible explanation for the observed poor knowledge in our study may be that there has been a general lack of community interventions to raise public awareness about antibiotics in India. The national awareness action plan on ABR in India started in 2017, a year after the data was collected [22]. Another explanation for poor knowledge can be the low level of education of a majority of our participants. This was further confirmed by the multivariable regression analysis, which showed an association between higher education and good knowledge (adjusted OR 24.16, 95% CI 9.77–59.73). This finding agrees with a study in Sweden where higher education was associated with a higher number of correct responses to the knowledge questions [33]. In addition, women were associated with good knowledge about antibiotics in the current study. Women might have received knowledge about antibiotics when taking care of family members during their sickness. However, men were more knowledgeable in a study conducted in Bangladesh [36]. This is an interesting result, which needs further probing to understand the reasons behind women having higher knowledge. The observed poorer knowledge in our study highlights the need to develop and implement community-focused interventional programs, e.g., educational programs about antibiotics, their action, use, safety, and resistance, to increase public awareness about antibiotic use and resistance.

Appropriate attitudes towards antibiotic use and ABR were documented in 40% of the study participants. This figure is lower than the results of the Kuwaiti study (59%) and an Indonesian survey on 573 subjects (68%) [25,35]. Patient expectations have been recognized as an important determinant for prescribing antibiotics at clinics, contributing to the overuse or misuse of antibiotics [31,37,38]. Despite having an overall poor attitude towards antibiotic use and resistance, more than 90% of our participants reflected that they would not pressurize healthcare workers to prescribe antibiotics. Prior studies have shown that many general practitioners felt pressure, mainly from the patients, to prescribe antibiotics, even if it is not indicated [39,40]. In our study, old age showed an association with a more inappropriate attitude to antibiotics [31,33]. Participants aged ≥ 65 years had 46% lower odds of appropriate attitude compared to those 18–40 years old (adjusted OR 0.54, 95% CI 0.35–0.86). Similar results were presented in a Swedish study and a Malaysian study, suggesting to develop interventions, focusing on elderly people to improve their attitude towards the use of antibiotics [31,33].

In our study, although women had a higher level of knowledge, they had a more inappropriate attitude towards antibiotics compared to the male participants. Previous studies showed inconsistent results regarding the association between gender and knowledge attitude about antibiotics [31,33]. In the Swedish study, men had better knowledge but a less appropriate attitude toward antibiotics, while in the Malaysian and Bangladeshi studies, men had overall good knowledge and a good attitude [24,33,36]. The differences could be due to varied social contexts and ongoing community health campaigns between countries and need to be studied in detail. Additionally, almost all participants (99%) in our study came from a family whose decision-maker was the father. The decision-maker of the family can influence the attitude and practices regarding the healthcare of family members. In our study, women had better knowledge but their attitude towards antibiotics might be influenced by the family decision-makers. Therefore, their attitude was less appropriate than those of the male participants.

Our study also highlighted that participants’ practice related to antibiotic use was significantly suboptimal (89% of participants). More than 80% of participants have bought antibiotics without a prescription (OTC). Similar results were shown in a previous study conducted among university students and their family members in Qatar, where 82% of respondents had used antibiotics without prescriptions (OTC) [41]. In India, it is known that antibiotics are accessible at retail pharmacies, and this represents a major contributor to antibiotic misuse and resistance [2]. This undesirable practice, observed in our study, reemphasizes the urgent need for better regulatory policies regarding the sale of antibiotics in community pharmacies in India. Moreover, in our study, the cost of a doctor’s visit was reported as the most common reason for buying antibiotics without a prescription. This suggests revising policies related to healthcare costs, making it more affordable for the population in lower income levels.

The practice of using leftover antibiotics among our participants was satisfactory (2% used leftover antibiotics) compared to the Malaysian, Swedish, and Kuwaiti studies (15%, 6%, and 30%, respectively) [25,31,33]. The most concerning issue of practice in our study was compliance with the prescribed treatment. The majority of participants (87%) stated that they would not take the entire course of prescribed antibiotics but rather stop when some symptomatic relief has been obtained. This result is two-fold higher than that in the Malaysian (46%) and Kuwaiti (33%) studies [25,31]. India launched the “Medicines with the Red Line” media campaign in early 2016 as an initiative to improve people’s knowledge and awareness about antibiotics and to stop antibiotic misuse [22], but 97% of participants in the current study reported not having noticed the “red line” at all. Our study was conducted a few months after the campaign’s launch, and thus, it is possible that the campaign had not reached a wide population. Therefore, its effectiveness and impact might not have been significant enough. This probably also supports the finding that most participants did not know whether they had taken antibiotics or not since they were unable to identify which drugs are antibiotics when obtaining drugs from retail pharmacies. Our finding suggests that evaluating people’s awareness of the red line campaign and its impact is necessary.

In addition, the multivariable regression analysis showed that factors such as higher education level, having a job, and living in urban areas, were associated with more appropriate antibiotic use practice. This is possible because people with a low level of education or unemployment, or living in rural areas generally, have poorer access to antibiotic-related or general healthcare-related information compared to those with higher education levels or living in the city [42]. Our finding suggests focusing interventions on unemployed people or those with low education levels or living in the countryside. Additionally, equitable access to healthcare services needs to be ensured.

In the present study, having good knowledge about URTI was positively and significantly associated with every domain in the KAP regarding antibiotic use and ABR. This finding suggests that people’s knowledge about common infections might be an important determinant for appropriate antibiotic use and ABR. Since URTIs are one of the most common infections in the community, a poor understanding of URTIs can cause antibiotic misuse, leading to resistance [17]. Importantly, the observed poor knowledge about URTI, and significantly suboptimal KAP about antibiotics among the attendees of the mass gathering in our study, raise the concern of infectious disease transmission associated with mass gatherings. Additionally, a study by Tim Bäckdahl et al. conducted among the pilgrims of the 2016 *Kumbh Mela* found that there were some important knowledge gaps about the transmission risk of tuberculosis [28], although these results do not directly show the health consequences of the mass gathering, but might act as a reminder of the potential mass gathering-associated public health risk. Future events, therefore, should adopt preventive measures to ensure the safety of the crowd, especially the risk of infectious disease transmission.

The poor KAP, and the existing inequalities in KAP between different socio-demographic groups observed in our study, suggest that interventions to promote good KAP are warranted and should be tailored to different social groups. For example, interventions should focus more on the elderly, men, and those with a low education level and who are unemployed. Furthermore, knowledge about common infections in the community, such as causes of the spread of infections, its treatment, and the role of antibiotics in its treatment, should also be integrated into the intervention plans. The intervention designs could be implemented via personal meetings, educational materials, and training campaigns. As stated in the Global Action Plan on Antimicrobial Resistance, these campaigns have been proven to improve general public awareness and the understanding of ABR and address its challenges [21]. Once the interventions are implemented, coordinated evaluations of these interventions, as conducted in the present study, are needed, and will provide feedback to policymakers to modify campaigns and enhance their effectiveness.

## 5. Strengths and Limitations

To the best of our knowledge, this is the first study conducted to explore and present the KAP regarding antibiotic use and ABR among the attendees of one of the largest mass gatherings in India. This could provide important insights into the prevention strategies for communicable diseases during mass gatherings in general. Another strength of the study is the large sample size and the diversity in participants’ socio-demographic characteristics. This could ensure the study’s precision and reliably statistically test the association between the KAP and socio-demographic characteristics, thereby drawing important conclusions. In addition, the study addressed participants’ knowledge about URTI, providing a complete picture of antibiotic use in communities in India.

The interpretation of the findings of this study, however, should be considered with certain potential limitations. First, in general, the cross-sectional design does not facilitate determining the causal relationship between KAP and the independent variables. However, this is the most feasible method of data collection in such settings. Second, the study sample is not a perfect representative of the Indian population due to the convenient sampling method and the fact that we only approached passing-by pilgrims who came to attend the mass gathering, which could affect the external validity of the findings. To amend this issue, we tried to collect data at different locations and time points of the event. A random sampling procedure could have produced a perfect study sample, but this was almost impossible due to the circumstances of the event, which included crowds of people constantly moving. Even though our results could not be generalized to the entire Indian population, they could serve as preliminary evidence of the general population’s KAP for developing relevant public health measures. Third, the data collection method used was the self-report, which made the findings subject to recall bias. Participants might have not recalled accurately their history of using antibiotics. Additionally, respondents might over-report or under-report some socially desirable attitudes or practices regarding antibiotic use. However, personal information (name, address, and phone number) was not collected and thus it is expected that respondents gave truthful answers. Finally, the instrument used to assess the KAP was not statistically validated and tested for reliability. However, translation, retranslation, and pilot tests were conducted prior to data collection.

## 6. Conclusions

In conclusion, the current study evidenced that the KAP regarding antibiotic use and resistance among the general population in India, was poor. Gender, occupation, age, and place of residence were factors that might affect the KAP. In addition, knowledge about URTI was closely associated with the population’s performance on KAP. Therefore, it is critical to design and implement multifaceted campaigns, considering socio-demographic characteristics, to promote the knowledge and awareness of the threat of ABR. The following contents ought to be included in the awareness campaigns: (a) common types of bacterial and viral infections and the role of antibiotics in treatment; (b) the emergence, causes, and consequences of ABR; and (c) how to prevent ABR, especially through patient compliance to the prescribed treatment.

## Figures and Tables

**Table 1 antibiotics-11-01473-t001:** Socio-demographic characteristics of the respondents (*N* = 1619).

Characteristics	*n* (%)
Age (mean ± SD)	39.0 ± 15.3
Age group (in years)	
18–40	1059 (65)
41–64	420 (26)
≥65	140 (9)
Sex	
Male	950 (59)
Female	669 (41)
Place of residence	
City	937 (58)
Village	682 (42)
Type of family	
Extended	1501 (93)
Unit	118 (7)
Education attainment	
Below high school	993 (61)
High school	430 (27)
Above high school	196 (12)
Occupation	
Have a job	1142 (71)
Do not have a job	477 (29)
Family’s decision-maker	
Father	1597 (99)
Mother	19 (1)
Grandparent	3 (0)

**Table 2 antibiotics-11-01473-t002:** Participants’ knowledge about URTI (*N* = 1868).

Question (Correct Responses)	Response	*n* (%)
1. What are the most typical symptoms of URTI? *(Running nose, sore throat, cough, fever)*	Correct	103 (6)
	Incorrect	1765 (94)
2. Is URTI a communicable disease? *(Yes)*	Correct	929 (50)
	Incorrect	939 (50)
3. What do you think are the main causes of URTI? *(Organisms or germs or contact with a sick person)*	Correct	439 (24)
	Incorrect	1429 (76)
4. Do you know what the causative organism of URTI is? *(Bacteria or virus)*	Correct	575 (31)
	Incorrect	1293 (69)
Total score	Median (IQR)	1 (0–2)
	0	611 (33)
	1	697 (37)
	2	354 (19)
	3	183 (10)
	4	23 (1)
Level of knowledge	Poor	1662 (89)
	Good	206 (11)

**Table 3 antibiotics-11-01473-t003:** Participants’ knowledge about antibiotic use and ABR (*N* = 1768).

Question (Correct answer)	Response	*n* (%)
1. What are antibiotics? (Medicines that *kill bacteria or prevent the growth of bacteria or prevent infection)*	Correct	122 (7)
	Incorrect	1646 (93)
2. Do antibiotics have side effects or adverse reactions? *(Yes)*	Correct	212 (12)
	Incorrect	1556 (88)
3. What can be the consequences of the irrational use of antibiotics? *(Antibiotic resistance, severe disease, problems in treatment in the future)*	Correct	192 (11)
	Incorrect	1576 (89)
4. Do you think that antibiotics are losing their efficacy? *(Yes)*	Correct	388 (22)
	Incorrect	1380 (78)
Knowledge score	Median (IQR)	0 (0–1)
	0	1124 (63)
	1	436 (25)
	2	159 (9)
	3	36 (2)
	4	13 (1)
Knowledge Group	Poor	1719 (97)
	Good	49 (3)

**Table 4 antibiotics-11-01473-t004:** Participants’ attitudes toward antibiotic use and resistance (*N* = 1662).

Question (Correct Answer)	Response	*n* (%)
1. Do you think that antibiotics should be kept as preserved medicines for the future? *(Yes)*	Appropriate	230 (14)
	Inappropriate	1432 (86)
2. If a doctor does not prescribe antibiotics, will you pressurize him/her to prescribe an antibiotic (s)? *(No)*	Appropriate	1611 (97)
	Inappropriate	51 (3)
3. If a doctor does not prescribe antibiotics, will you go to another doctor who might prescribe an antibiotic (s)? *(No)*	Appropriate	1544 (93)
	Inappropriate	118 (7)
4. Who is responsible for the irrational use of antibiotics? (*Allopathic doctors, local non-allopathic practitioners, patients, medical shops/pharmacies, pharmaceutical companies, unlicensed doctors)*	Appropriate	606 (37)
	Inappropriate	1056 (63)
Attitude score	Median (IQR)	2 (2–3)
	0	6 (1)
	1	129 (8)
	2	870 (52)
	3	506 (30)
	4	151 (9)
Attitude Group	Inappropriate	1005 (60)
	Appropriate	657 (40)

**Table 5 antibiotics-11-01473-t005:** Participants’ practice regarding antibiotic use (*N* = 1911).

Question	Response	*n* (%)
1. Have you ever purchased and administered an antibiotic without a doctor’s prescription? *(No)*	Appropriate	369 (19)
	Inappropriate	1542 (81)
2. Have you noticed a Redline on some medicine strips? *(Yes)*	Appropriate	48 (3)
	Inappropriate	1863 (97)
3. Would you use any leftover antibiotics or the same prescription to purchase the antibiotics whenever you have similar symptoms with previous infections? *(No)*	Appropriate	1869 (98)
	Inappropriate	42 (2)
4. Do you purchase the complete antibiotic(s) course written in the prescription by the doctor? *(Yes)*	Appropriate	909 (48)
	Inappropriate	1002 (52)
5. Do you comply to the entire course of the prescribed antibiotics (compliance to administer prescribed antibiotic(s) for complete duration)? *(Yes)*	Appropriate	247 (13)
	Inappropriate	1664 (87)
Practice score	Median (IQR)	2 (1–2)
	0	15 (1)
	1	583 (30)
	2	1091 (57)
	3	208 (11)
	4	13 (1)
	5	0 (0.0)
Practice Group	Inappropriate	1690 (89)
	Appropriate	221 (11)

**Table 6 antibiotics-11-01473-t006:** Multivariable logistic regression analysis on factors associated with participants’ KAP about antibiotic use and resistance.

Factors	Adjusted OR (95%CI)		
	Knowledge	Attitude	Practice
Sex (female vs. male)	**2.26 (1.05–4.85)**	**0.65 (0.49–0.86)**	1.26 (0.86–1.87)
Age group (vs. 18–40 years old)			
41–64 years old	0.97 (0.41–2.27)	**0.72 (0.55–0.95)**	0.99 (0.67–1.48)
≥65 years old	NA	**0.54 (0.35–0.86)**	1.23 (0.65–2.32)
Education (vs. below high school)			
High school	2.18 (0.74–6.40)	1.12 (0.85–1.47)	1.16 (0.79–1.69)
Above high school	**24.16 (9.77–59.73)**	1.17 (0.80–1.71)	**1.92 (1.22–3.02)**
Occupation (having vs. not having a job)	1.35 (0.62–2.95)	1.12 (0.81–1.50)	**1.80 (1.17–2.78)**
Family type (extended vs. unit)	**0.32 (0.14–0.72)**	**0.51 (0.32–0.83)**	0.86 (0.50–1.51)
Place of residence (city vs. village)	1.07 (0.52–2.21)	**1.28 (1.01–1.63)**	**1.56 (1.11–2.20)**
Knowledge about URTI (good vs. poor)	**2.90 (1.35–6.22)**	**5.85 (3.94–8.68)**	**3.56 (2.40–5.28)**

Values in bold represent statistically significant values.

## Data Availability

All data generated or analyzed during this study are included in the present article. The datasets contain the personal information of the participants, therefore, are not available publicly. However, the data are available through the corresponding author on reasonable request.

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
