# Peer review of "Knowledge, Attitude, and Practice Regarding Antibiotic Use and Resistance for Upper Respiratory Tract Infections among the Population Attending a Mass Gathering in Central India: A Cross-Sectional Study"

_antibiotics, 2022, doi:10.3390/antibiotics11111473_

Round 1
Reviewer 1 Report
Well written paper and like the way authors presented their hypothesis and conclude results.
Can be accepted as it is.
Author Response
Reviewer 1
Well written paper and like the way authors presented their hypothesis and conclude results.
Can be accepted as it is.
Response: Thank you for your kind and encouraging words.

Reviewer 2 Report
Thank you for the invitation to review this manuscript. I have a few concerns about this draft;
1. The authors did not provide any information on the validity and reliability. What was the reliability assessment measure? was it Cronbach alpha? the method of translation is not clear
2. Please upload the data collection form as an annexure, as I did not receive any annexure file during this round of review. The data collection form in Hindi, as well as English, will help the researchers.
3. What was the sampling method? Was it convenient?
4. Please elaborate on the data collection form in more detail in the method section. How many questions were present in each section of the data collection form and on which scale the responses were measured?
5. The scoring system should be described in a separate heading of "outcome variables" or "outcomes"
6. how the question "1. What are the three typical symptoms of URTI? " Can provide accurate information on the knowledge of participants as the definition of typical varies across the population? For me, the typical symptoms also include malaise or fever. A runny nose is not always common in RTIs.
7. Please provide more details on the definitions of knowledge, attitude, and practice groups in the method section.
8. The odds ratio for family type is less than 1 for knowledge and attitude, it is not clear that joint or single family is related to better knowledge or attitude. please define the reference variable. For knowledge, it seems that single-family is associated with good knowledge. How were the logistic regression analyses performed? What were the dependent and independent variables?
Please provide details on how the participants were approached and selected. Please provide an estimated figure on how many people attended this event and from which part of India they usually belonged.
Reviewer 3 Report
This is a very interesting cross sectional study on the factors associated with behaviors that have an impact on the development of antibiotic resistance in India. ABR is of course a topic that remains of great and increasing interest, in both High and Low-Middle Income Countries.
Still I think some changes could be necessary:
- Abstract
o I would write the total number of participants only in the Results section, and not in the Methods section, since it was not pre-determined
- Introduction
o Population’s knowledge, perception and attitude are crucial in many public health issues, and the current pressing covid-19 pandemic is no exception. I would suggest to remind this and the covid-19 pandemic, for example in:
§ Genovese C, La Fauci V, Di Pietro A, Trimarchi G, Odone A, Casuccio A, Costantino C, Restivo V, Fantini M, Gori D, Azara A, Deiana G, Castaldi S, Righi E, Palandri L, Panciroli G, Bianco A, Licata F, Cosentino S, Mistretta A, Marranzano M, Ragusa R, Gabutti G, Stefanati A, Prato R, Fortunato F, Martinelli D, Icardi G, Panatto D, Amicizia D, Fabiani L, Moretti A, Di Risio D, Siliquini R, Voglino G, Bert F, Lorini C, Bonaccorsi G, Torre I, Pennino F, Pavia M, Di Giuseppe G, Paolantonio A, Villari P, Marzuillo C, Messina G, Rivieri C, Nante N, Majori S, Tardivo S, Moretti F, D'Amato S, Mazzitelli F, Giunta I, Lo Giudice D, Pantò G, Signorelli C, Squeri R. COVID-19: opinions and behavior of Italian general population during the first epidemic phase. Acta Biomed. 2022 Jul 1;93(3):e2022262. doi: 10.23750/abm.v93i3.12262. PMID: 35775780; PMCID: PMC9335423.
o I would probably leave the statements about the potential risks of large mass gatherings in the Discussion section and remove this part from the Introduction and Methods sections, since those gathering are a premise/context of a study, but those potential risks were not part of what was evaluated
- Methods
o 2.2 The authors state that the majority of the questions were in multiple choice format. I would suggest to report how many of those questions were in that format, and the available options in the paper for those questions, maybe in a separate Appendix
o 2.3 How was the value of p calculated? I would suggest to report its value and citations are needed as to “previous literature”
- Results
o 3.1 Shouldn’t 1915/2065 be 92,7% instead of 92,3% ?
o Table 1.
§ It is stated in the manuscript that socio-demographic characteristics were available in only 1619 out of 1915 participants, but age group is available in all of them. Since the total of the age groups is different for the other characteristics I would signal that, or use a different table
§ Type of family is separated between “joint” and “single”. I would understand that joint family is a family in which male descendants live with their parents together with their own families, is this correct? I would suggest maybe to explain the meaning of “joint family” in the manuscript, for better clarity to audiences in which this type of family is not very common.
o Table 6
§ I would indicate in the table caption that values in bold represent statistically significant values, for better clarity
§ I would report aOR values that “go in the same direction” with the 95%IC interval all either above or below 1. For example, for the factors associated with better knowledge about antibiotic resistance, sex, education and knowledge about URTI are all above 1, while family type is below 1.
- Strenghts and limitations
o The Authors extend their study to the general population in India. I would add to the discussion some elements about the possible selection bias in using a population of only pilgrims in their study and why they state that bias does not apply in their study.
Author Response
Reviewer 3: This reviewer has suggested some good points so you can follow those
This is a very interesting cross sectional study on the factors associated with behaviors that have an impact on the development of antibiotic resistance in India. ABR is of course a topic that remains of great and increasing interest, in both High and Low-Middle Income Countries.
Still I think some changes could be necessary:
- Abstract
o I would write the total number of participants only in the Results section, and not in the Methods section, since it was not pre-determined.
Response: Thank you for your useful suggestion. We have revised the abstract accordingly and removed the total number of participants in the Methods part.
- Introduction
o Population’s knowledge, perception and attitude are crucial in many public health issues, and the current pressing covid-19 pandemic is no exception. I would suggest to remind this and the covid-19 pandemic, for example in:…
Response: Thank you for this valuable suggestion. We have now included a few sentences about this point in the Introduction and suggested a reference as well.
"The population’s knowledge, perception, and attitudes are crucial in several public health issues. The progression of Covid-19, a respiratory tract infection, as a pandemic is the most relevant example where the lack of knowledge about how to prevent the spread of infection was one of the major factors that claimed several lives (7). Moreover, the improved knowledge played a crucial role in minimizing the spread as well. It is noteworthy that the consequences of overuse of antibiotics during the Covid-19 pandemic period are yet to be studied."
- I would probably leave the statements about the potential risks of large mass gatherings in the Discussion section and remove this part from the Introduction and Methods sections, since those gathering are a premise/context of a study, but those potential risks were not part of what was evaluated.
Response: Thank you for this suggestion. We have shortened the description of the mass gathering in the introduction and Heading 2.1. Study design and population in the method part.
- Methods
o 2.2 The authors state that the majority of the questions were in multiple choice format. I would suggest to report how many of those questions were in that format, and the available options in the paper for those questions, maybe in a separate Appendix.
Response: Thank you. We have now provided information on the proportion of multiple-choice questions in our questionnaire (under heading 2.2). We have also attached the questionnaire as an annexure to this submission.
"A majority of questions (28/30 questions) were in multiple-choice format to overcome time constraints during the mass gathering. Two open-ended questions were: year of birth and the definition of antibiotics."
- 3 How was the value of p calculated? I would suggest to report its value and citations are needed as to “previous literature”.
Response: Thank you for pointing this out. We have added the information on how we selected the proportion p for sample size estimation (heading 2.4 Statistical analysis in the method section). We chose the p as 50% since we used a different instrument to assess KAP compared to previous literature, which also used different tools, and we had three different proportions for three outcomes of KAP.
"As previous studies have used different instruments to assess the KAP and we had three proportions of interest, we decided to use 0.5 as the p to maximize our sample size."
- Results
o 3.1 Shouldn’t 1915/2065 be 92,7% instead of 92,3%?
Response: Thank you for pointing this out. It was a typing mistake. We have corrected it (92.7%) in the revised manuscript.
o Table 1.
- It is stated in the manuscript that socio-demographic characteristics were available in only 1619 out of 1915 participants, but age group is available in all of them. Since the total of the age groups is different for the other characteristics I would signal that, or use a different table
Response: Thank you. We have corrected the age group information (Table 1). We are now only using the information on 1619 participants with complete data on all socio-demographic characteristics to report age group data to make it consistent with the other characteristics, as well as to be in line with our method of handling missing values stated in the method section.
"Complete data on socio-demographics were available for 1619 participants."
- Type of family is separated between “joint” and “single”. I would understand that joint family is a family in which male descendants live with their parents together with their own families, is this correct? I would suggest maybe to explain the meaning of “joint family” in the manuscript, for better clarity to audiences in which this type of family is not very common.
Response: Thank you. Following the reviewer's comment, we have changed the terms regarding this variable to improve its clarity for the audience. We are now using "extended family" (previously termed as a joint family) and "unit family" (previously termed as single-family). The description for these terms is shown in the corresponding question in the questionnaire (Annexure I). Joint family or extended family refers to a family with more than two generations (for example grandparents, uncles' family, and cousins and with or without their children living together in a household). Single-family or unit family refers to a family consisting of a married couple with/without their children. The description for these terms is shown in the corresponding question in the questionnaire (Annexure I).
o Table 6
- I would indicate in the table caption that values in bold represent statistically significant values, for better clarity
Response: Thank you. We have added the caption on bold values to Table 6.
"*Values in bold represent statistically significant values."
- I would report aOR values that “go in the same direction” with the 95%IC interval all either above or below 1. For example, for the factors associated with better knowledge about antibiotic resistance, sex, education and knowledge about URTI are all above 1, while family type is below 1.
Response: Thank you for this valuable suggestion. We have revised our result description accordingly (heading 3.6. Factors Associated with KAP About Antibiotic Use and ABR).
"Regarding knowledge, females (adjusted OR - aOR 2.26, 95%CI 1.05-4.85), above high school education level (aOR 24.16, 95%CI 9.77-59.73), and better URTI knowledge (aOR 2.90, 95%CI 1.35-6.22) were significantly associated with better knowledge about anti-biotics, whereas an extended family (aOR 0.32, 95%CI 0.14-0.72) was associated with poor knowledge. No association was found for age, occupation, and place of residence. For attitudes, age group 41-64 (aOR 0.72, 95%CI 0.55-0.95), age group ³65 (aOR 0.54, 95%CI 0.35-0.86), female (aOR 0.65, 95%CI 0.49-0.86), and extended family (aOR 0.51, 95%CI 0.32-0.83) were associated with lower odds of appropriate attitudes. However, living in the city (aOR 1.28, 95%CI 1.01-1.63) and better knowledge about URTI (aOR 5.85, 95%CI 3.94-8.68) were linked to appropriate attitudes."
- Strengths and limitations
o The Authors extend their study to the general population in India. I would add to the discussion some elements about the possible selection bias in using a population of only pilgrims in their study and why they state that bias does not apply in their study.
Response: Thank you. Following the reviewer's comment, we have revised our discussion on strengths and limitations. We have removed the strength that we said our results could provide evidence for the general population in India, and we now have acknowledged the limitation of potentially limited generalizability of our findings to the entire Indian population due to convenient sampling and the fact that our study was conducted on only pilgrims attending a mass gathering.
"... This could provide important insights into the prevention strategies for communicable diseases during mass gatherings in general."
"... and the fact that we only approached passing by pilgrims who came to attend the mass gathering,"
" Finally, the instrument used to assess KAP was not statistically validated and tested for reliability. However, translation, retranslation, and pilot tests were conducted prior to data collection."

Round 2
Reviewer 2 Report
Thanks for clarification.